# Tunnel FET and MOSFET Hybrid Integrated 9T SRAM with Data-Aware Write Technique for Ultra-Low Power Applications

Wenjuan Lu [1], Yixiao Lu [1], Lanzhi Dong [1], Chunyu Peng [1,2], Xiulong Wu [1,*], Zhiting Lin [1] and Junning Chen [1]

[1] School of Integrated Circuits, Anhui University, Hefei 230601, China
[2] The Key Laboratory of Intelligent Computing & Signal Processing (Anhui University), Ministry of Education, Hefei 230601, China
* Correspondence: xiulong@ahu.edu.cn

**Abstract:** In this paper, a Tunnel FETs (TFETs) and MOSFETs hybrid integrated 9T SRAM (HI-9T) with data-aware write technique is proposed. This structure solves the problem of excessive static power consumption caused by forward p-i-n current in the conventional 7T TFET SRAM (CV-7T), and the problem of weakened writing ability caused by the use of the TFET-stacked structure of the most advanced combined access 10T TFET SRAM (CA-10T). The simulation results demonstrate that the static power consumption of HI-9T is reduced by three orders of magnitude compared with CV-7T at a 0.6 V supply voltage and the ability to maintain data is more stable. Compared with CA-10T, the write margin (WM) of HI-9T is increased by about 2.4 times and the write latency is reduced by 54.8% at 0.5 V supply voltage. HI-9T still has good writing ability under the 0.6 V supply voltage and the CA-10T cannot write normally. Therefore, HI-9T has good overall performance and is more advantageous in ultra-low power applications.

**Keywords:** TFET; forward p-i-n current; SRAM; ultra-low static power consumption

## 1. Introduction

Low-power design is very important to SRAM, which acts as the core component of System-on-Chip (SoC) and occupies most of the area and power consumption [1,2]. With the development of manufacturing technology, the prominent short channel effect of MOSFET can lead to excessive leakage current, and further increases the static power consumption of SRAM. Although static power consumption can be effectively reduced by scaling down supply voltage (VDD), the sub-threshold swing (SS) of MOSFET is hard to break through the limit of 60 mV/dec at room temperature, which makes it extremely challenging [3,4]. Compared to conventional MOSFETs, TFETs with asymmetric source and drain design, higher current switching ratio, smaller turn-off current, and steeper SS (less than 60 mV/dec) due to the band tunneling effect are the most promising candidates to replace CMOS in ultra-low power applications [5–10]. However, the asymmetric structure along with the distinct current transport mechanism of TFET also results in unidirectional conduction and forward current. Figure 1 illustrates the $I_{DS}$-$V_{DS}$ output characteristics of an n-TFET with positive and negative $V_{DS}$ based on the GaN/InN TFET model available from Notre Dame University [11]. As shown in Figure 1, When $V_{DS} > 0$ V, the p-i-n diode operates in reverse bias, and the conduction is controlled by the gate voltage ($V_g$). When $V_{DS} < 0$ V, the p-i-n diode is forward biased and shows a conduction current (i.e., forward p-i-n current) only barely controlled by the gate terminal [12]. In particular, the uncontrollable forward p-i-n current grows rapidly with the $V_{SD}$ increasing, which will limit its application in circuits [13]. Many studies have been conducted on the TFET applications in SRAM design [13–23], but a series of problems are still not resolved, especially the following two problems: 1. The problem of large static power consumption caused by forward p-i-n current in the transmission transistor. 2. In order to solve the problem of the forward p-i-n current, the writing ability of the transmission transistor

using the TFET stack structure is greatly reduced. This paper presents a new HI-9T SRAM with the following advantages: 1. The influence of the forward p-i-n current is solved by using MOSFET as the transmission transistor. 2. The source and drain of the transmission transistor have no direct grounding path, which solves the problem of the large static leakage power consumption of MOSFET under low power supply voltage. 3. The structure of the writing operation completed by the combination of NTFET and MOSFET solves the problem that the writing ability is greatly weakened due to the stacking of TFET.

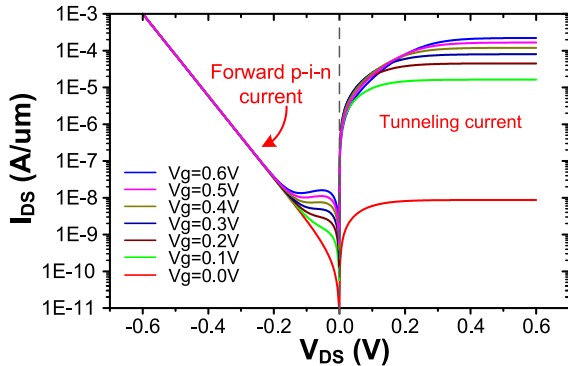

**Figure 1.** Current transfer characteristics of 20 nm GaN/InN TFET on logarithmic scale.

The remainder of the paper is organized as follows. Section 2 elaborates the influences of the forward p-i-n current and the stacked TFETs. Section 3 introduces the proposed HI-9T SRAM structure and explains its basic operation principles in detail. The comparison of metrics and simulation results obtained are presented in Section 4. Section 5 concludes the paper.

## 2. The Influences of the Forward P-I-N Current and the Stacked TFETs

Next, we will introduce the influence of the forward p-i-n current and the problems of the improved structure from the two basic TFET cell structures, the inward facing FT-6T cell structure (IFT-6T) and outward facing FT-6T cell structure (OFT-6T). According to Refs. [12,13,15,16], there is a fundamental tradeoff between readability and write ability. In the general case, IFT-6T SRAM cell, as shown in Figure 2a, can read data well, but the writing function of it cannot be well realized. Meanwhile, the OFT-6T, as shown in Figure 2b, can smoothly realize the writing function, but the read delay is very large or even unacceptable, especially at an ultra-low voltage. This is due to the conduction current from the bitline-bar (BLB) to the storage node QB is not a normal conduction current but a forward p-i-n current during the reading operation of OFT-6T, and the voltage of BLB is further affected by the magnitude of the forward p-i-n current. As shown in Figure 2b, assuming that Q and QB are '0' and '1', respectively, the bit line (BL) is precharged to a high level, and the word line (WL) is set to a high level during the read operation. The source voltage of the access transistor N3 is greater than the drain voltage, so there will be an uncontrolled forward p-i-n current from BL to Q, resulting in a slow decrease in BL voltage. Similarly, when OFT-6T at hold state, although the word line (WL) is low, there is still a forward p-i-n current in the access transistor N3. Although the forward p-i-n current is small at ultra-low voltage, it will further affect the static power consumption and stability of OFT-6T. Moreover, reducing the bit line voltage is also a better solution to decrease the forward p-i-n current, but it will increase the complexity of the power management circuit, and consideration should be given to the effect of the reduced bitline voltage on other SRAM cells in the array, i.e., the half-selection problem [12].

Replacing OFT-6T access transistors with MOSFET can overcome the problem caused by the unidirectional conduction of TFET and eliminate the influence of the forward p-i-n current on cell power consumption, but this structure cannot fully take the advantage of the low cutoff current of TFET. Because if Q is '0', as shown in Figure 2b, and supposing

N3, N4 are MOSFET transistors, there will be a leakage from the bit line through the MOSFET N3 to Q, and then through N1 to the ground, where N1 is turned on. Although the asymmetric 6T TFET SRAM [13] with write assistant technology can improve the writing ability of it, the influence of the forward p-i-n current on the circuit is not taken into account.

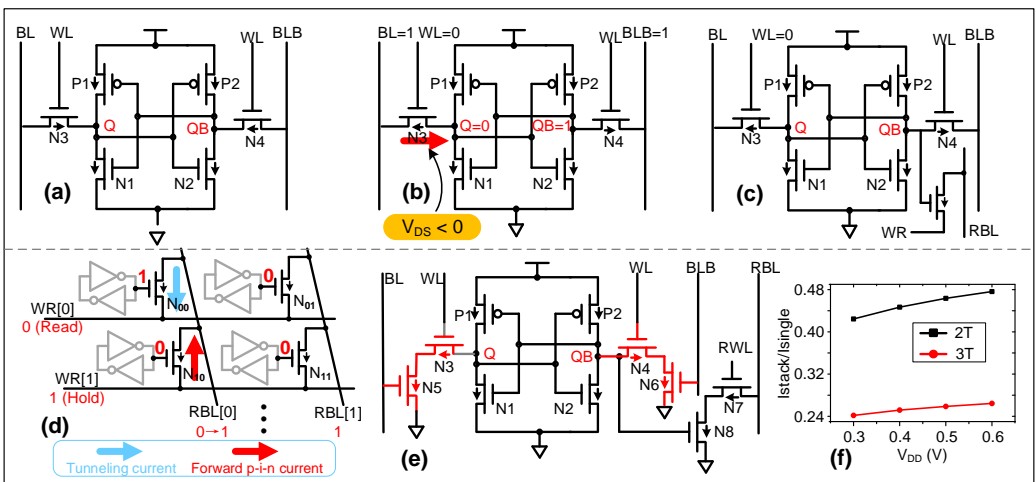

**Figure 2.** (**a**) IFT-6T. (**b**) OFT-6T with forward p-i-n current. (**c**) CV-7T. (**d**) The forward p-i-n current problem encountered by the CV-7T in the array during the read operation. (**e**) CA-10T. (**f**) The ratio of the on-current when two and three TFETs are stacked to that of a single TFET.

Furthermore, Ref. [15] adds an additional TFET transistor to implement the read function on the basis of the OFT-6T structure, forming the CV-7T SRAM cell structure, as shown in Figure 2c. Although this read–write separation structure can well overcome the influence of TFET unidirectional conduction on the read-write function, it does not eliminate the forward p-i-n current. On the issue of the forward p-i-n current, the structural part of the CV-7T's write function will encounter the same situation as OFT-6T in the hold and write status. Moreover, the problem caused by forward p-i-n current is even worse during read operation in array. As shown in Figure 2d, RBL[0] is precharged to '1', when WR[0] is set to '0', RBL[0] will be discharged to low level through $N_{00}$. However, due to the WR[1] being set to '1', the forward p-i-n current of $N_{10}$ will slow down the discharge speed of RBL[0]. If there are a lot of similar states as that of $N_{10}$ on RBL[0], the read speed and accuracy of SRAM will be further damaged.

In addition, the TFET and MOSFET hybrid SRAM have also been researched [17,24,25]. The Ref. [17] have presented a mixed TFET-MOSFET 8T SRAM cell to improve the HSNM. However, the cross coupled inverter pair of the cell uses MOSFETs, so the advantages of TFETs are not sufficiently utilized in terms of low power consumption. Similarly, the SRAM based on the Schmitt trigger [18,19] structure can improve the read/write noise margin of the cell, but its access transistors also have a forward p-i-n current.

Recently, as shown in Figure 2e, Ref. [20] proposes a new combinational access 10T TFET SRAM (CA-10T), which can eliminate the impact of the forward p-i-n current and obtain higher read and hold static noise margin (SNM) and lower power consumption. However, due to the fact that stacked TFETs can seriously weaken the transistor's conduction ability and affect the logic function of the circuit [26], the combined transmission mode will result in a weak write ability. In order to show the deterioration in the conduction ability, Figure 2f presents the ratio of the on-current when two and three TFETs are stacked to that of a single TFET. As is shown, with the voltage varying from 0.3 V to 0.6 V, the on-current when two and three TFETs are stacked are reduced by about 58% to 52% and by 76% to 73%, respectively, compared to that of a single TFET. Thus, stacking TFET will seriously weaken its conduction ability, which will affect the write ability of SRAM. Therefore, in order to improve the conduction capacity of the stacked TFETs and improve

the write capacity of the CA-10T SRAM, it is necessary to additionally increase the size of the stacked TFET device, which will increase its layout area.

In view of the above problems, a HI-9T SRAM cell structure with a data-aware write technique for ultra-low power application is proposed in this paper. The proposed structure is simulated by the GaN/InN TFET model of Notre Dame University [11] and the 65 nm MOSFET model of TSMC in cadence. In this work the width size of MOSFET and TFET transistors is set to 120 nm, the length sizes of TFET and MOSFET are set to 20 nm and 60 nm, respectively.

## 3. Proposed HI-9T with Data-Aware Write Technique

The proposed HI-9T is shown in Figure 3a, where the access transistors, N6 and N7, are implemented by MOSFETs, and the others are implemented by TFETs. In detail, the drains of N6 and N7 are connected to the storage nodes Q and QB, respectively. Meanwhile, their sources are electrically connected to the drain of N3. Owing to the $V_{DS}$ of N3, as well as N5, being not less than '0', thus, the proposed circuit is free from the forward p-i-n current. The basic operation principles and the half-selected issue are detailed below.

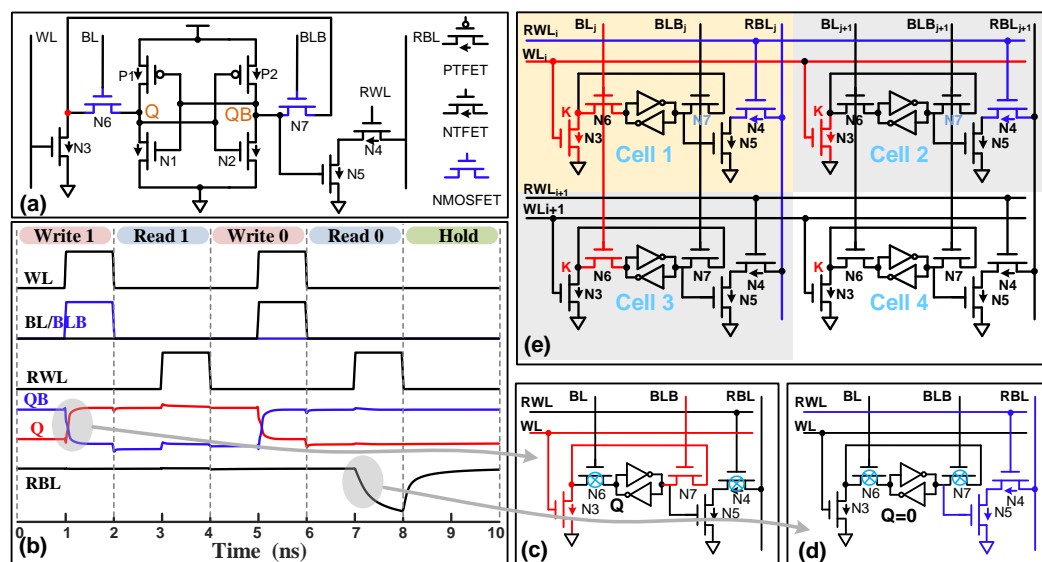

**Figure 3.** The schematic (the blue device represents NMOS), simulation waveforms, write '1', read '0' and 2 × 2 array for half-select disturb analysis of the proposed HI-9T are shown in (**a**–**e**), respectively.

### 3.1. Write, Read and Hold Operation

Figure 3b gives the simulation waveforms of the proposed HI-9T. During the write operation, WL is set to '1', and the write operation will be accomplished with the help of the data-aware write technique, that is the write path will be determined by the data to be written, according to Ref. [27].

As shown in Figure 3c, if BL and BLB are driven by the data as '0' and '1', respectively, N3 and N7 will be activated. Then, QB will be discharged to low levels through N7 and N3 transistors, and Q will be feedback to high levels. As a result, Q and QB will be written as '1' and '0', respectively. On the contrary, Q and QB will be written as '0' and '1', respectively. Here, in order to enhance the writing capability of HI-9T, the gate voltages of N6 and N7 are powered by an independent power supply corresponding to the standard operating voltage of the MOSFET device, rather than a 0.6 V array power supply. For the read operation, as shown in Figure 3d, the read–write separation structure similar to CA-10T is used in the proposed schematic. If QB is '1', the precharged RBL will be discharged by N4 and N5 when RWL is activated. Instead, it still remains high voltage. During the hold mode, WL, BL and BLB will be set as '0'. Moreover, since the two leakage paths of the storage nodes all pass through TFET N3, the proposed HI-9T SRAM can take full

advantages of the ultra-low cutoff current of TFET and achieve low power consumption in the hold state.

### 3.2. Half-Selected Issue

In order to interpret the half-selected disturb issue, which is very important to SRAM design, a 2 × 2 memory array using the proposed HI-9T structure is given in Figure 3e. As shown in Figure 3e, if cell 1 is selected for writing '0', $WL_i$ is set to '1', $BL_j$ and $BLB_j$ are set to '1' and '0', respectively. However, if N3 of cell 2 is activated, it is free of half-select disturb issue owing to the off-state N6 and N7 with the $BL_{j+1}$ and $BLB_{j+1}$ keeping the low voltage level. Meanwhile, due to the $WL_{i+1}$ and $BLB_j$ are in low voltage level, N3 and N7 of cell 3 are off, and the awaking $BL_j$ does not cause a half-select problem for cell 3, similarly. Additionally, the read path of the proposed HI-9T is decoupled from the internal storage node, thus, the storage node will not be disturbed during the read operation.

## 4. Comparison and Discussion
### 4.1. Write Static Noise Margin

According to the analysis in Section 2, the stacked TFET structure of CA-10T weakens the write ability of SRAM owing to the reduction in the conduction current. In order to overcome this problem, here, a hybrid stacking method of TFET and MOSFET is proposed to improve its conduction capability. Figure 4a gives the schematic of two stacked methods. After simulation, Figure 4b shows the ratio of the on-current of a 2T stacked TFETs and the on-current of a hybrid stack structure to the on-current of a single TFET under different operating voltages. Under the working voltage of 0.3 V to 0.6 V, the simulation results validate that the on-current of the hybrid stacked structure will be greatly improved compared to the 2T stacked TFET. The on-current of the hybrid stack structure is only about 10% to 20% lower than that of a single TFET, which can improve the writing ability of HI-9T SRAM effectively.

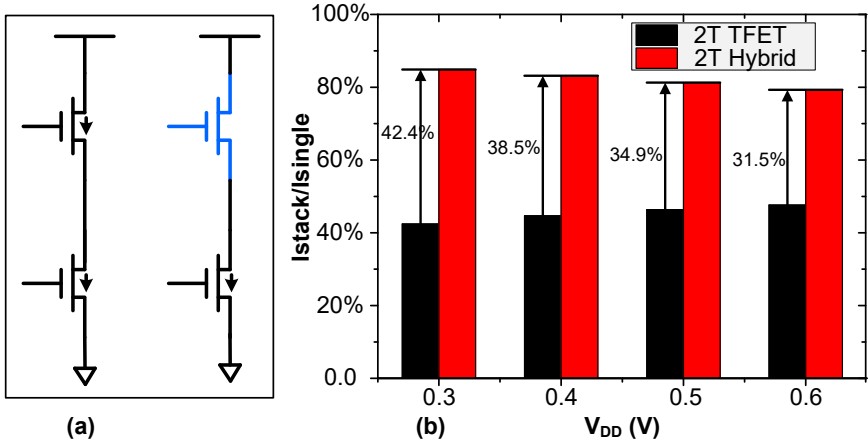

**Figure 4.** (**a**) Two TFETs-stacked structure (**left**), TFET and MOSFET hybrid stacked structure (**right**). (**b**) The proportion of current of stacked TFETs to that of single TFET.

Figure 5a shows the simulation results of the CV-7T, CA-10T and proposed HI-9T in terms of WM with different supply voltages. By the way, WM is measured by WL and defined as the margin between VDD and WL when the storage node reaches the switching voltage of the corresponding half-cell. It is shown that HI-9T and CV-7T have similar WM, but the WM of HI-9T achieves about 61~138% improvement compared with that of CA-10T with the supply voltage varying from 0.3 V to 0.5 V. In particular, under 0.6 V supply voltage the CA-10T cannot be written in the data successfully, as Figure 5b shows, because of the stacked TFETs. Whereas, the WM of the proposed HI-9T still reaches 346 mV.

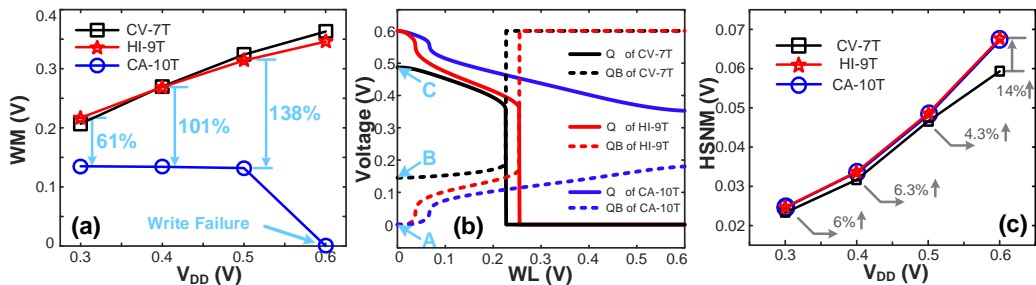

**Figure 5.** (**a**) Writing Margin of Three Structures under Different Supply Voltages (**b**) Writing Operation Waveforms of Three Structures under 0.6 V Supply Voltage (**c**) Hold Static Noise Margin of Three Structures under Different Supply Voltages.

Meanwhile, this paper also provides the voltage variation of the two storage nodes and the current of related access transistors and pull-up transistors when the SRAM cell writes normally at 0.6 V supply voltage, which is shown in Figure 6. From the Figure 6a,b, the current flowing through the access transistor N3 is always smaller than that of the pull-up transistor P1 when WL is activated, which means that the CA-10T cell cannot complete the write operation consequently. Meanhile, from the Figure 6c,d, during the writing operation of the HI-9T cell, the current flowing through the access transistor N6 is larger than that of the pull-up transistor P1 at the initial stage of the jump of Q from '1' to '0', which ensures the successful write operation and larger WM, accordingly.

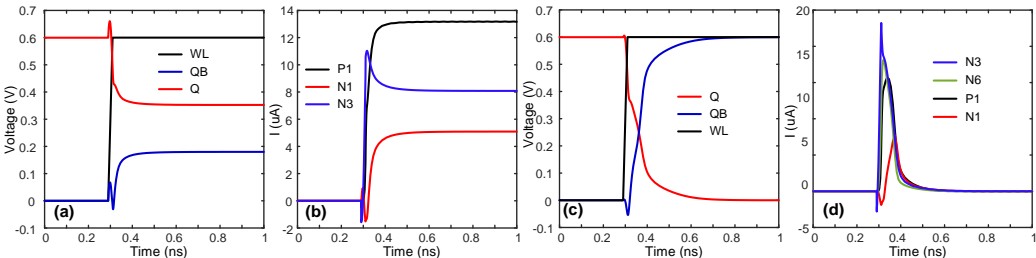

**Figure 6.** The voltage and current curves during the write operation. Where, (**a**,**b**) are those of CA-10T, and (**c**,**d**) are those of HI-9T.

### 4.2. Hold/Read Static Noise Margin

The hold and read static noise margin (HSNM and RSNM) are also key parameters to evaluate the SRAM performance. Here, assume that the Q and QB of the above mentioned cells are initialized to '1' and '0', respectively.

In the hold state, WL is set as '0', and the simulation results of the HSNM comparison among CV-7T, CA-10T and HI-9T structures are shown in Figure 5c. When WL = '0', on account of the forward p-i-n current of the access transistors, the two storage nodes (Q and QB) of the CV-7T structure cannot be maintained at the VDD and ground, which corresponds to the point C and B in Figure 5b. On the contrary, the CA-10T and proposed HI-9T are free of this situation, as shown by the point A in Figure 5b. The HSNM of the proposed HI-9T is similar as that of CA-10T, and it has about 14% improvement compared with CV-7T under the 0.6 V supply voltage, as Figure 5c shows. In addition, although the hold SNM of HI-9T is only slightly higher than that of CV-7T from the data point of view, it has made great progress, because the hold SNM of the three kinds of SRAM themselves is relatively small. This is due to the fact that the delayed saturation in the TFET device results in large cross-over region/current between the n-type and p-type devices in TFET inverter, which degrades the sharpness of the voltage transfer characteristic (VTC) of the TFET inverter, and the TFET inverter stability in TFET SRAM cell [17,18]. As shown in Figure 7a, when the input voltage changes, the output voltage of the inverter composed of the NTFET and PTFET is obviously not as steep as the voltage output by the CMOS

inverter; thus, the HSNM value of the TFET SRAM measured by the butterfly curve is small, as shown in Figure 7b.

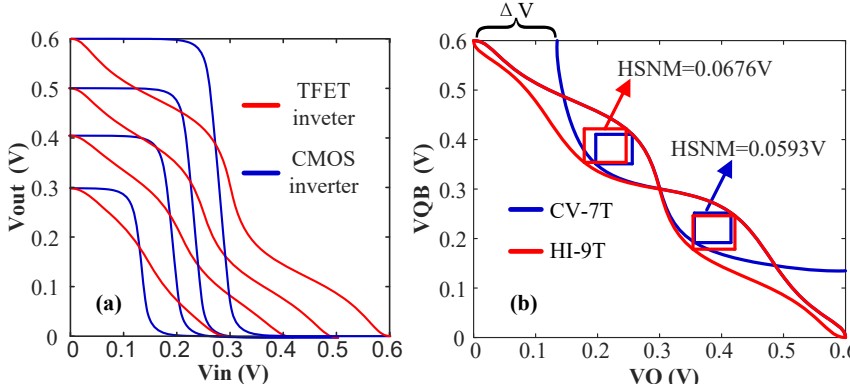

**Figure 7.** (**a**) Output characteristics of TFET inverter and CMOS inverter. (**b**) The HSNM comparison of CV-7T and HI-9T with butterfly curve.

Furthermore, the stability of circuit is a critical factor that must be considered in the design of SRAM cell. The stability of the cell can be influenced by PVT, and this part mainly considers the influence of process volatility. The fluctuation of the process causes a change in the threshold voltage of the transistor, where we assume that the threshold voltage fluctuation of the transistor satisfies the Gaussian distribution [28] and its standard deviation is set to 20 mV. Figure 8a–c show 1000 Monte Carlo simulations of HSNMs at 0.6 V operating voltage for these three SRAM cells, and the Figure 8d lists the mean values and the standard deviations of HSNMs to make a contrast. It can be seen that the HSNM mean values of HI-9T and CA-10T SRAM without forward p-i-n current are large than that of CV-7T, and the standard deviation is more than 38% lower than CV-7T. Therefore, HI-9T has strong anti-process volatility and stability. Moreover, since the three SRAM cells compared are all separate read and write structures, their RSNM and HSNM are the same, so there is no separate comparison of read margin .

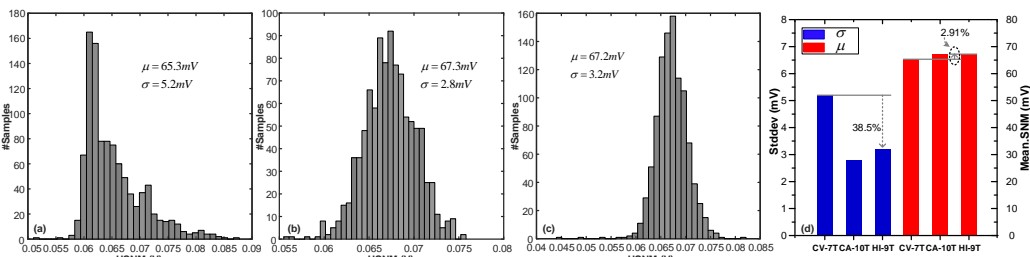

**Figure 8.** HSNM histogram of (**a**) CV-7T, (**b**) CA-10T, (**c**) Proposed HI-9T, (**d**) Stddev ($\sigma$) and mean-value ($\mu$) of SNM @VDD = 0.6 V.

### 4.3. Power Consumption

Simultaneously, the static power consumption of the above three SRAM cells is compared, and the results are shown in Figure 9a. It indicates that the static power consumption of the proposed HI-9T and CA-10T is reduced by three orders of magnitude, compared with that of CV-7T at a 0.6 V supply voltage, respectively. This further verifies that eliminating the forward p-i-n current can greatly reduce the static power consumption of the circuit. Meanwhile, the dynamic power consumption for performing a write and read operation is analyzed at a frequency of 500 MHz with a 10 fF load capacitor of RBL, as shown in Figure 9b. Due to the forward p-i-n, the current issue is conducive to write operations for CV-7T; therefore, the switching time of it is relatively small. As a result, it has the lowest dynamic power consumption. Meanwhile, the dynamic power consumption

of CA-10T is the largest, with the long time of conduction caused by the low conduction current of the stacked TFETs.

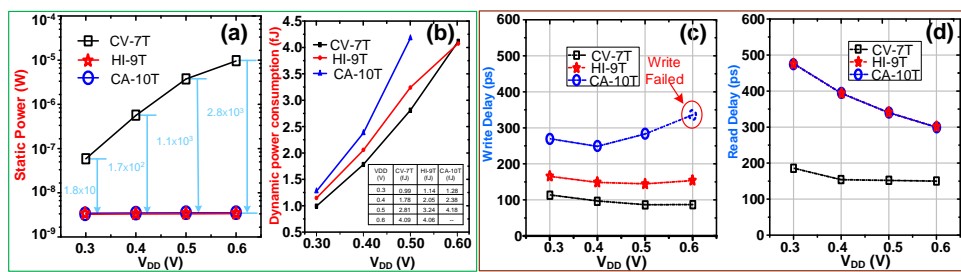

**Figure 9.** (**a**) Static Power Consumption (**b**) Dynamic Power Consumption (**c**) Write Delay (**d**) Read Delay of Three Structures under Different Supply Voltages.

*4.4. Read/Write Delay*

Read-write latency is an important indicator of memory performance. Figure 9c,d shows the write and read delays of three kinds of SRAM cells, respectively. As shown in Figure 9c, because of the low conduction current of stacked TFETs in the CA-10T structure, the presented HI-9T SRAM obtains less write delay and higher write speed, comparatively. Additionally, the read delay of both HI-9T and CA-10T structures is larger than that of the CV-7T, which is shown in Figure 9d, but the advantages of the HI-9T structure are focused on lower power consumption, larger HSNM, higher writing speed and better read stability.

**5. Conclusions**

In this paper, a new HI-9T structure is proposed, which mainly solves the following three problems: the problem of excessive static power consumption caused by the forward p-i-n current, the problem of large static leakage power consumption caused by the use of MOSFET as the transmission transistor, and the problem of using the TFET stacking structure, which leads to a significant decrease in writing ability. The proposed structure and comparison structures are simulated by the cadence and key performance comparison, which is summarized in Table 1, as follows: compared with CV-7T, the static power consumption of HI-9T is reduced by 3 orders of magnitude at 0.6 V supply voltage . The Monte Carlo simulation of the hold static noise margin shows that HI-9T has better and more stable data retention ability than CV-7T. Compared with CA-10T, the write margin of HI-9T is increased by about 2.4 times and the write delay is reduced by 54.8% at 0.5 V supply voltage. HI-9T also has a write margin of 346 mV for the 0.6 V operating voltage, which CA-10T cannot write normally. Therefore, the HI-9T proposed in this paper has great advantages in low power SRAM-integrated applications.

**Table 1.** Comprehensive performance comparison table.

| VDD | HSNM (mV) | | | | WM (mV) | | | | Write Delay (ps) | | | | Read Delay (ps) | | | | Static Power (nW) | | | | Dynamic Power Consumption (fJ) | | | |
|---|---|---|---|---|---|---|---|---|---|---|---|---|---|---|---|---|---|---|---|---|---|---|---|---|
| | 0.3 V | 0.4 V | 0.5 V | 0.6 V | 0.3 V | 0.4 V | 0.5 V | 0.6 V | 0.3 V | 0.4 V | 0.5 V | 0.6 V | 0.3 V | 0.4 V | 0.5 V | 0.6 V | 0.3 V | 0.4 V | 0.5 V | 0.6 V | 0.3 V | 0.4 V | 0.5 V | 0.6 V |
| **CV-7T** | 23.2 | 31.7 | 46.6 | 59.5 | 207 | 271 | 325 | 363 | 113 | 97 | 89 | 87 | 183 | 156 | 153 | 150 | 90 | $8.5 \times 10^2$ | $5.5 \times 10^3$ | $1.4 \times 10^4$ | 0.99 | 1.78 | 2.81 | 4.09 |
| **CA-10T** | 24.8 | 33.7 | 48.3 | 67.5 | 135 | 133 | 131 | – | 260 | 250 | 283 | – | 473 | 394 | 341 | 302 | 5 | 5 | 5 | 5 | 1.28 | 2.38 | 4.18 | – |
| **HI-9T** | 24.6 | 33.7 | 48.7 | 67.9 | 217 | 269 | 314 | 346 | 167 | 148 | 128 | 162 | 473 | 394 | 341 | 302 | 5 | 5 | 5 | 5 | 1.14 | 2.05 | 3.24 | 4.06 |

**Author Contributions:** Conceptualization, W.L., Y.L. and L.D.; validation, Y.L. and L.D.; data curation, C.P. and Z.L.; writing—original draft preparation, Y.L. and L.D.; writing—review and editing, Y.L. and W.L.; supervision, W.L. and X.W.; project administration, J.C.; funding acquisition, W.L. and X.W. All authors have read and agreed to the published version of the manuscript.

**Funding:** This work was supported in part by the National Key R&D Program of China under Grant 2018YFB2202803, the National Natural Science Foundation of China under Grant 62104001, Natural Science Foundation of Anhui Province under Grant 2008085QF322, Natural Science Foundation of the Higher Education Institutions of Anhui Province under Grant KJ2019A0031, and the Doctoral Research Funding Project of Anhui University under Grant Y040418177.

**Data Availability Statement:** All data included in this study are available upon request by contacting with the corresponding author.

**Conflicts of Interest:** The authors declare no conflict of interest.

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
