# Peer review of "Tunnel FET and MOSFET Hybrid Integrated 9T SRAM with Data-Aware Write Technique for Ultra-Low Power Applications"

_electronics, doi:10.3390/electronics11203392_

Round 1
Reviewer 1 Report
1) English must be improved. There are some persistent errors, like the beginning of paragraphs with ". And ...". Avoid "can´t" and replace by "cannot". Try to reduce the use of "the", like in line 124, 129, etc ...
Please discuss the use of cascode (?) instead of stacked (for example in line 36). Is "stacked" usually used in the literature for this field? Also, avoid qualitative terms like superior (line 38) and similars.
2) Some figures are too small (2, 3 and 5). They must be increased in size in order to get a more comprehensive analyses by the reader.
Fig. 5b should be divided for Q and QB cases, in order to understand the difference for each solution (CV7, CA or HI). Also, please indicate if yy axes in this figure is the supply voltage VDD.
3) The structure of the paper must be improved and some paragraphs must be rewritten.
- Please consider the need for the discussion in section 2, regarding the pin forward current issue, since these problem seems to be negligeble for this work, as also noted by the paragraph of line 118); further, the proposal does not present any results for negative supply voltages, like depicted in figures 5a), 5c), 7 and 9. Please explain.
- Please explain in more detail the meaning of the setence lines 129-131, it is confusing.
- From the read of Fig. 7a) transfer characteristic I cannot find out inverter functionality in the red TFET inverter, since the characteristic is almost linear. Can you explain this? In this case, the FET/TFET stack is not an option but a necessity.
- It is not clear for the reader that the HI solution is the very best when compared with CA and CV counterparts; in fact, in some specs CV-7T is a better option. Please discuss this in more detail in section 4 and in the conclusions.
4) Finally, since this is a proposal based on simulations, without experimental results, I was expecting at least some PVT corners in order to evaluate if the HI proposal is functional in a large number of cases. The authors refer corners (line 201) but do not present any simulation results. Please explain this absence.
Reviewer 2 Report
1. An interesting paper regarding TFETs and MOSFETs hybrid integrated 9T SRAM for ultra-low power application.
2. What is the target process of the proposed SRAM? What is the Hspice model used in comparison?
3. Is there any tape-out plan? I hope that author can provide the silicon measurement data in the future.
4. I suggest that author can provide the performance comparison table to highlight the power or other important performance indices.
5. Does TFETs and MOSFETs hybrid design style increase the difficult and cost of chip manufacture?
Reviewer 3 Report
The paper needs improvement, the following corrections are necessary:
1… The abstract must be modified, adding the most important things found in the investigation, including the most interesting results and conclusions.
2… The problem statement should be improved, showing the weaknesses of previous research and showing the benefits of this research
3… The statement of the problem and the objectives of the research are not clear.
4… Please show one by one the contributions of the paper.
5… If possible, I suggest making some experimental measurements with an oscilloscope.
6… The conclusions should be improved, with respect to the objectives and contributions of the paper, finally the conclusions must be consistent with the results obtained.
7… Bibliographic references must be improved and updated, several recent papers from 2017 to 2021 must be cited, more than half of the bibliographic citations are very old.
Round 2
Reviewer 1 Report
Still some "the" must be eliminated.
Even if MCarlo simulations seems to validate the proposal, I do not understand why not including corners simulations discussion in the work, since it is a simulation (not experimental) based paper.
Reviewer 3 Report
The corrections were made correctly, thank you very much